# Exploring Caloric Restriction in Inpatients with Eating Disorders: Cross-Sectional and Longitudinal Associations with Body Dissatisfaction, Body Avoidance, Clinical Factors, and Psychopathology

**DOI:** 10.3390/nu15153409

**Published:** 2023-07-31

**Authors:** Matteo Martini, Paola Longo, Tiziano Tamarin, Federica Toppino, Annalisa Brustolin, Giovanni Abbate-Daga, Matteo Panero

**Affiliations:** Eating Disorders Center, Department of Neuroscience “Rita Levi Montalcini”, University of Turin, Via Cherasco 11, 10126 Turin, Italy; matteo.martini@unito.it (M.M.); paola.longo@unito.it (P.L.); tiziano.tamarin@unito.it (T.T.); federica.toppino@unito.it (F.T.); annalisa.brustolin@gmail.com (A.B.); matteo.panero@unito.it (M.P.)

**Keywords:** body image, dietary restriction, weight suppression, binge eating

## Abstract

Reduction in food intake is an important feature of eating disorders (EDs). However, whereas self-reported cognitive control over food (i.e., dietary restraint) is commonly assessed, we are not aware of any study evaluating the actual reduction in caloric intake (i.e., caloric restriction, CR) and its relationships with psychopathological, clinical, and anamnestic factors in individuals with EDs. In this study, we quantified caloric intake, CR, and weight suppression in 225 ED inpatients and explored significant relationships with self-reported eating symptoms, body dissatisfaction, body avoidance, personality, and affective symptoms. For underweight inpatients (n = 192), baseline predictors of caloric intake and restriction at discharge were assessed through a data-driven approach. CR at admission was significantly related to eating symptomatology, state anxiety, and body image. In regression models, CR, higher BMI, binge-purging symptoms, and the interaction between weight suppression and CR were significantly related to body dissatisfaction. The best psychopathological predictors of caloric intake and restriction at discharge for underweight inpatients were perfectionistic concern over mistakes and state anxiety. These results suggest that caloric restriction is associated to relevant ED features and warrant for a multidimensional assessment of ED psychopathology.

## 1. Introduction

Caloric restriction (CR) refers to the reduction in food intake to create a negative energy balance and is used as a therapeutic instrument for individuals with obesity [1]. Usually, a CR of around 25% (i.e., 25% reduction in caloric intake from the calculated daily calorie need) is employed. CR in the same range is also supposed to have health benefits for individuals with normal weight [1,2]. CR is different from dietary restraint in that the latter refers to the cognitive control exerted over food intake but does not necessarily involve actual caloric reduction [1]. Whereas dietary restraint has been associated with body image concerns and the development of disordered eating, recent studies challenged the belief that the actual controlled reduction in food intake during CR programs would be associated with dysregulated eating and body image concerns [3]. It is important to note, however, that individuals with any psychiatric issue were excluded from studies on CR [4], and different considerations should be made for individuals who are affected by eating disorders (EDs) or are predisposed to their development.

In the context of EDs, reduction in food intake is an unhealthy method for losing body mass and maintaining often dangerously low weight. Chronic restriction is common to different EDs, albeit usually most sustained in anorexia nervosa (AN). Most commonly, hospitalization for EDs occurs when low body weight and eating restrictions become difficult to manage in the outpatient setting or when medical and/or psychiatric comorbidities require urgent treatment [5]. In this context, clinicians must focus the treatment of the affected individual on restoring bodily parameters and improving underlying psychopathological conditions [6]. Hallmarks of ED psychopathology are concerns about weight and body shape, as well as high levels of anxiety; in fact, the correlation between an increase in anxiety and restrictions in patients with EDs is known [7,8]. Furthermore, individuals who engage in restrictive eating are also characterized by high cognitive control and perfectionism, which are currently understood as traits favoring the onset of EDs, as well as disorder-maintaining mechanisms [9,10].

To our knowledge, no study has assessed the degree of CR (i.e., the actual reduction in food intake, different from the self-reported dietary restraint) in a population of ED inpatients and its relationship with body dissatisfaction and avoidance, psychopathological scores, and hospitalization treatment outcomes. As recently pointed out by authors in the ED field [10], the relationships linking the psychopathological, dietary, and anamnestic aspects of caloric intake should be assessed in more detail in order to develop better treatment strategies for individuals with ED. Whereas body mass index (BMI) is routinely calculated, represents an element for diagnostic classifications, and has been associated with a plethora of clinical and psychopathological variables, other measurable features such as negative energy balance, body composition, and weight suppression have been much less investigated but could represent relevant elements for understanding the connections between physical and cognitive features of EDs. For instance, weight suppression (i.e., the difference between the lifetime highest and the current BMI of an individual) has been shown to be associated with the severity of shape and weight concerns and emerged as useful in some studies as a predictor and moderator of treatment outcomes [11,12,13,14,15], but its role across the ED spectrum is still not clear and most likely complex [16]. Assessing weight suppression together with CR could help in understanding its role as a predictor or moderator of ED psychopathology and treatment outcomes.

Furthermore, whereas reductions in self-reported dietary restraint are commonly reported after inpatient treatment, we are not aware of any study assessing the relationships between admission variables and actual caloric intake at discharge from an inpatient ED program. Regarding the underweight ED population (i.e., mostly individuals with a diagnosis of AN), an increase in BMI is the best predictor of positive outcomes [17]. However, discharge from the inpatient hospital service often occurs at low BMIs, especially after emergency hospitalization [18]. Since treatment is most of the time voluntary, the caloric intake reached at discharge could represent an indicator of compliance with treatment and of further improvement after discharge. Understanding what baseline factors are associated with caloric intake and restriction at discharge could shed light on important factors for the outcome of hospitalization.

Despite the sheer quantity of calories introduced, diet composition could both impact and be influenced by health and psychopathology in individuals with EDs. Furthermore, cooking methods influence nutritional qualities and the content of potentially dangerous compounds present in foods, such as advanced glycation end-products (AGEs), which are associated with metabolic disease and could be involved in gut microbiome dysfunction [19,20]. For instance, even though 10–30% of AGEs introduced with foods are absorbed, a diet rich in AGEs could be involved in maintaining obesity [20]. On the opposite weight polarity, due to AN psychopathology, the diet of individuals with AN is characterized by products generally considered as healthy and with low levels of AGEs (i.e., low-fat foods, foods cooked at low temperatures, and raw foods) [19,21]. Nevertheless, paradoxically higher plasma AGEs levels have been found in individuals with AN in comparison to controls, suggesting increased endogenous production of these compounds, coupled with concomitant oxidative dysfunction [22]. Given the conceptualization of EDs as metabo-psychiatric disorders [23], the increasing importance recognized in the ED field to the role of the gut microbiome [24], immunity [25], and inflammation [26], more attention should be given to the nutritional and metabolic effects of foods chosen by patients and introduced during renutrition [24,27]. Associations between diet composition and depressive and anxiety symptoms have emerged from the literature [28]; however, similar studies in the ED field are still lacking. If significant associations between caloric restriction in EDs and clinical factors emerge, future studies should evaluate the effects of nutritional composition, metabolic, and inflammatory actions of food ingested in this relation.

With this study, we were interested in exploring the following questions: (1) Is the percentage of CR specifically associated with body image measures and psychopathology in a population of inpatients with EDs? (2) Does the current body mass and the amount of weight lost over the course of an individual’s life have a role in these relationships? (3) Lastly, what are the clinical and psychopathological predictors influencing caloric intake reached after an inpatient treatment program?

In accordance with these questions, we formulated the following aims:(1)Provide an assessment of the degree of caloric restriction (CR) in a population of individuals with ED at their first access to inpatient treatment, and investigate the associations between CR and a comprehensive battery of psychopathological variables.(2)Explore the relationship linking body image questionnaires, CR, weight suppression, and other clinical variables.(3)Explore the predictors of caloric intake and restriction at the end of inpatient treatment for underweight individuals.

We expected CR, weight suppression, and their interactions to be significantly related to body image measures. Furthermore, we expected higher body dissatisfaction, avoidance, and anxiety levels at admission to emerge as significant predictors of CR at discharge.

All the analyses presented in this study are exploratory in nature.

## 2. Materials and Methods

### 2.1. Participants

We consecutively recruited individuals with a diagnosis of ED admitted to inpatient treatment at the Eating Disorder Center of the University Hospital of Turin. Inpatient treatment in this specialized unit is delivered according to international guidelines [6], and treatment costs are covered by the National Health System. The access to inpatient treatment is often through the emergency room due to the acute worsening of clinical conditions; however, it can also be planned when there are difficulties in the treatment at the outpatient level. During the hospital stay, a multidisciplinary equipe follows the inpatient, and individualized goals in terms of nutritional and clinical care are agreed. Psychological sessions are offered in order to help the patient understand the possible causes of the worsening of symptoms, as well as intervening on the maintaining factors for the disorder.

Exclusion criteria for this study were age <18 years old, presence of psychotic disorders, and presence of active alcohol, cannabis, hallucinogens, inhalants, opioids, sedatives, cocaine, amphetamines, or methamphetamine use disorder. Only individuals with complete questionnaires on eating symptoms and body image were included.

The number of individuals initially included was 250. The number of individuals who completed the questionnaires was 225. The number of underweight individuals in this sample was 192.

The study was conducted in accordance with the Declaration of Helsinki and approved by the Ethics Committee of the University Hospital of Turin (protocol code 0073951, date of approval 9 July 2021).

### 2.2. Questionnaires

Eating symptoms were assessed with the widely used Eating Disorders Examination Questionnaire (EDE-Q), Italian version [29]. Cronbach’s alpha in the sample was 0.95.

Body image was assessed with the Body Shape Questionnaire (BSQ), Italian version [30], and the Body Image Avoidance Questionnaire (BIAQ), Italian version [31]. Both questionnaires were previously used with patients with EDs [32], and BSQ was used in the CALERIE study [3]. Cronbach’s alpha in the sample was 0.97 for BSQ and 0.88 for BIAQ.

Affective symptoms were assessed with the State Trait Anxiety Inventory (STAI), Italian version [33], and Beck Depression Inventory (BDI), Italian version [34]. Cronbach’s alpha in the sample was 0.96 for STAI and 0.87 for BDI.

The Frost Multidimensional Perfectionism Scale (FMPS), Italian version [35], was used to assess perfectionism. Cronbach’s alpha in the sample was 0.93.

### 2.3. Procedure

Clinical and sociodemographic variables were collected at admission. Diagnosis was assessed via clinical interview [36]. Height and weight were measured. The highest weight reached during the course of the individual’s life was assessed through self-report. Caloric intake at admission was evaluated during the initial assessment with a dietitian who calculated an estimate of the daily calories contained in the diet reported by the patient in the previous month. Caloric intake at discharge represents the amount of daily calories reached after agreed gradual increments during the inpatient stay. Physical activity in the month prior to the admission was quantified by the clinician through the patient’s self-report, and recorded as hours per week in which the individual engaged in physical exercise, sports, or intense walking. Physical activity at discharge reflected the amount of activity observed by the clinician and clinical staff during the inpatient stay. The level of physical activity at admission and discharge was then coded as sedentary, light activity (corresponding to up to 3 days/week), moderate activity (corresponding to up to 5 days per week), or strong activity (corresponding to up to every day of the week).

These variables allowed calculating BMI (kg/m^2^), kcal/kg, weight suppression, daily calorie need, and degree of CR.

Studying CR in individuals with ED presents some challenges. Reviewing the literature, we found no consensus on the best formula for basal metabolic rate (BMR) calculation in underweight individuals [37,38,39,40]. Previous studies found that the measured BMR in underweight individuals is generally lower than BMR derived from commonly used equations. However, evidence showed an acceptable performance of Mifflin–St Jeor equations [41]. These represent an approximation, but it is not the aim of this study to precisely assess BMR in ED patients.

In order to calculate daily calorie need, BMR was first calculated using the Mifflin–St Jeor equations [42]:

For female individuals,
10 × weight (kg) + 6.25 × height (cm) − 5 × age (years) − 16;(1)

For male individuals,
10 × weight (kg) + 6.25 × height (cm) − 5 × age (years) + 5.(2)

Then, BMR was multiplied by a coefficient accounting for the level of physical activity: 1.200 for sedentary, 1.375 for light activity, 1.550 for moderate activity, and 1.725 for strong activity.

The percentage CR was calculated as follows:100 × (daily calorie need − caloric intake)/daily calorie need.(3)

Weight suppression was calculated by subtracting the current BMI from the highest BMI [12]. In the case when current BMI was the highest ever reached, then weight suppression was 0.

Caloric intake expressed in kcal/kg, weight suppression, daily calorie need, and CR were calculated at admission and discharge for underweight individuals.

### 2.4. Data Analysis

To explore the associations between CR and clinical and psychopathological variables (aim 1), a correlational analysis between caloric intake expressed in kcal/kg, CR, BMI, weight suppression and psychopathological questionnaires was conducted. The significance level was set at *p* < 0.05. Holm p-value correction for multiple comparisons was applied.

The aims were explored through a combination of machine learning approaches and traditional statistics. For aim 2, we were interested in modeling the relationship linking body image questionnaires, CR, weight suppression, and their interaction, while accounting for duration of illness, age, BMI, and the presence of binge-purging symptoms. Upon visual inspection, we noticed that BSQ total score appeared to have a bimodal distribution. To investigate whether a mixture model would be more appropriate than a linear regression, we used the function stepFlexmix of the R package flexmix to identify the optimal number of components in the model, which was selected through the lower BIC. As can be seen in the Appendix A, BIC was very close between one and two components, but lower for the one-component model. Therefore, we proceeded as planned with a linear regression using BSQ total score as the dependent variable, and introducing CR, weight suppression, duration of illness, age, BMI, binge-purging symptoms, and the interaction between CR and weight suppression as predictors. A second regression model was then run with BIAQ total score and the same predictors.

For aim 3, we followed the method suggested by Fife and D’Onofrio [43], which consists of running random forest models to identify the best predictors of a variable in the data, and then inserting them in a linear regression model. This practice allows combining data-driven insights with traditional and more easily communicable statistics. Firstly, we ran a random forest with all the psychopathological and clinical variables contained in the first two tables. Then, we ran a linear regression inserting the variables—we chose a priori to use the first five—with the highest variable importance as predictors. We performed this process both for caloric intake expressed in kcal/kg and for CR at discharge since they could provide different information.

To allow for an interpretation of effects, all continuous predictors in the regression models were scaled and mean-centered so that one standard deviation in the predictor corresponded to the amount indicated by the regression coefficient in the dependent variable when all other predictors were held constant. All regression models were checked through the comprehensive model assessment provided by the performance R package.

All the analyses were run in R version 4.3.0 [44] using RStudio. Main packages used were easystats [45], party [46], flexmix [47], flexplot [48], and tidyverse [49].

## 3. Results

### 3.1. Caloric Restriction and Correlations

The mean daily caloric intake in the whole sample was 19 kcal/kg, whereas mean BMR and daily calorie need accounting for level of physical activity were respectively 1299 and 1779 kcal/day (Table 1). Figure 1 shows these variables across the different BMI groups. Application of the formula for CR resulted in a mean percentage CR of 54 (Table 1). Questionnaire scores for the whole sample are reported in Table 1.

The mean CR in the underweight sample was 56% at admission and 6% at discharge (Table 2). Mean weight (kg) gained per week was 0.38 kg/sett, and there was a significant albeit small increase of mean BMI from admission to discharge (Table 2).

Regarding the correlational analysis, after controlling for multiple comparisons, kcal/kg at admission was negatively related to BSQ total score (r(223) = −0.31, *p* < 0.001), BIAQ social activities (r(223) = −0.3, *p* = 0.001), BIAQ eating-related control behavior (r(223) = −0.37, *p* < 0.001), BIAQ total score (r(223) = −0.32, *p* < 0.001), EDE-Q eating restraint (r(223) = −0.29, *p* = 0.001), eating concern (r(223) = −0.27, *p* = 0.004), shape concern (r(223) = −0.27, *p* = 0.004), weight concern (r(223) = −0.23, *p* = 0.48), global score (r(223) = −0.29, *p* = 0.001), and to STAI state-anxiety (r(217) = −0.26, *p* = 0.011). Lower caloric intake corresponded to greater scores in the questionnaires. CR accounting for activity levels negatively related to BMI at admission (r(223) = −0.35, *p* < 0.001) and positively related to BSQ total score (r(223) = 0.26, *p* = 0.007), BIAQ social activities (r(223) = 0.3, *p* < 0.001), BIAQ eating-related control behavior (r(223) = 0.37, *p* < 0.001), BIAQ total score (r(223) = 0.3, *p* = 0.001), EDE-Q eating restraint (r(223) = 0.33, *p* < 0.001), shape concern (r(223) = 0.25, *p* = 0.017), global score (r(223) = 0.28, *p* = 0.002), and STAI state-anxiety (r(217) = 0.25, *p* = 0.025). Higher restriction corresponded to greater scores in the questionnaires.

Weight suppression showed no significant association with clinical and psychopathological variables.

All correlations are reported in the Appendix A.

### 3.2. Body Dissatisfaction, Caloric Restriction, and Weight Suppression

In the first regression model, CR, BMI, binge-purging symptoms, and the interaction between CR and weight suppression resulted significantly associated with BSQ total scores (Table 3). All predictors were positively associated except for the interaction term.

Except for binge-purging symptoms, the second regression model showed the same significant associations of the previous model for BIAQ total scores (Table 4).

In both models, multicollinearity was low (i.e., VIF below 5; Appendix A).

Figure 2 shows the relationship between BSQ and BIAQ total scores and caloric restriction for different BMI groups, and Figure 3 shows the moderation effects of higher weight suppression levels.

### 3.3. Predictors of Caloric Intake/Restriction at End of Inpatient Treatment

Running the random forest, the variables with the highest variable importance in relation to caloric intake at discharge were admission BMR, length of stay (days), BMI, daily calorie need, and FMPS concern over mistakes. Variables with the highest variable importance in relation to CR at discharge were daily calorie need, FMPS concern over mistakes, STAI state anxiety, admission BMR, and EDE-Q global score.

All variable importance values in random forest are reported in the Appendix A.

The results of the linear regression with caloric intake at discharge as dependent variable are reported in Table 5. BMR, length of stay, and FMPS concern over mistakes were significant predictors. BMR and FMPS concern over mistakes were negatively related to the dependent variable, whereas length of stay had positive effects.

The results of the linear regression with CR at discharge as dependent variable are reported in Table 6. Daily calorie need, FMPS concern over mistakes, and STAI state anxiety were all significantly and positively related to the dependent variable.

In both models, multicollinearity was low (i.e., VIF below 5; Appendix A).

## 4. Discussion

In this study, we aimed to assess the degree of caloric restriction (CR) in inpatients with eating disorders (ED) and evaluate its relationships with body image, psychopathological, and clinical variables. Furthermore, we were interested in evaluating the baseline predictors of caloric intake and restriction at the end of inpatient treatment for underweight individuals.

Caloric intake (expressed as kcal/kg) and the percentage CR at admission were, respectively, negatively and positively correlated with body image, eating symptoms, and state anxiety. Exploring more in detail the significant associations of body image measures in regression models, we found that body dissatisfaction as measured by BSQ was significantly associated with a higher degree of CR, a higher BMI, the presence of binge-purging symptoms, and a negative association with the interaction between weight suppression and CR. Analogous results were found for body image avoidance, with the exception of binge-purging symptoms. Through a data-driven approach, FMPS concern over mistakes emerged as an important psychopathological predictor of both caloric intake and restriction at discharge from treatment.

These results suggest that quantitatively assessed restrictive behaviors are significantly related to body image concerns in an ED population; however, the improvement in these behaviors during inpatient treatment is negatively influenced in significant measure by cognitive control and affective symptoms [10,50]. These observations are in line with the need for a multidimensional evaluation and treatment of EDs and with the importance of studying nutritional and psychopathological variables together in order to unravel their complex interactions.

The mean CR in our sample was over 50%. Even though for underweight individuals, the use of a formula for basal metabolic rate could have led to overestimations, this value is well above the range used in treatments with CR, thus limiting the possibility of comparisons with studies involving non-ED obese and normal-weight individuals. In our clinical sample, in line with our expectations, CR emerged as relevantly related to ED features across the weight and ED spectrum. Regarding the correlational analysis, CR was significantly and positively related to EDE-Q eating restraint and BIAQ eating-related control behavior. This result suggests that these subscales correctly reflect the degree of actual eating restrictions in inpatients, thus indicating that, in these individuals, reported cognitive restraint is consistent with actual food reduction. The other significant correlations regarded body image measures and state anxiety, which represent other important hallmarks of ED psychopathology.

Analyzing the relationship between body image measures and clinical variables in regression models, we found that body dissatisfaction in inpatients is more pronounced for individuals with a higher BMI, who engage in binge-purging behaviors, and who display higher CR. This finding could be interpreted in the sense that, in some individuals with binge-purging symptoms, body dissatisfaction and clinical symptoms can reach levels warranting inpatient treatment, even at a BMI that would be manageable in the outpatient setting. Furthermore, the presence of binge-purging symptoms interferes with pure CR, and this could represent an element of frustration for these individuals. Interestingly, the expected association between CR and more severe body dissatisfaction and avoidance was independent from BMI (i.e., present across the weight spectrum). These findings suggest that the percentage CR at admission to inpatient treatment could complement weight-related indicators such as BMI, whose connection to psychopathology is limited [51].

The relationship with actual caloric restriction, however, appears to be less strong for those individuals for whom the difference between their current and highest lifetime BMI is higher. This finding could mean that, for individuals with EDs, having a BMI that is much lower in comparison to their highest could moderate the restrictions induced by body dissatisfaction. Alternatively, in these individuals, body dissatisfaction and avoidance could induce less eating restriction in comparison to individuals who have lost less weight over the course of their lifetime. These results are in line with previous evidence suggesting the role of weight suppression as a moderator of body shape concerns across the weight spectrum [52]. The mean weight suppression in our study was comparable to other studies [12]. However, in our sample, weight suppression considered individually showed neither significant correlations with the baseline measures nor significant effects in the regression models. Furthermore, weight suppression did not emerge as a relevant predictor of caloric intake and restriction and discharge for underweight inpatients. In summary, even though weight suppression has shown interesting associations and is an easy-to-calculate indicator, results from the current and previous studies still do not precisely define its role in ED psychopathology and treatment trajectories [16]. It is likely that long-term evaluations are needed to fully grasp its influences on weight gain and symptom modification across time. In the short to medium term, indicators related to the current symptomatology, such as the degree of restriction and physical activity, seem to better correlate to the clinical picture.

The analyses conducted at the end of inpatient stay for underweight individuals highlight the relevance of perfectionistic concern over mistakes as a baseline predictor associated with both lower caloric intake reached and higher levels of caloric restriction still present at discharge. Similarly, baseline anxiety symptoms were associated with higher caloric restriction at discharge, whereas neither specific eating symptoms nor admission or discharge BMI showed prominent roles in these associations. These results are in line with previous evidence underlying the central role of maladaptive perfectionism [9,53,54,55] and affective traits in the maintenance of ED symptomatology [8,50,56]. It can be argued that for individuals who present to treatment with a high level of perfectionism and/or anxiety symptoms, prioritizing the work on these symptoms could be more effective than psychological work only focused on body shape and weight concerns [57]. The baseline assessment should be comprehensive in order to provide the optimal treatment delivered to each individual [58]. Nonetheless, bidirectional associations between caloric restriction and psychopathology in individuals with EDs should be considered and assessed in future studies, with a focus both on quantitative restrictions and the nutritional content of foods.

As far as we are aware, this is one of the first studies to quantitatively assess eating restrictions in a population of ED inpatients and evaluate their relationship with clinical, psychopathological, and body image measures. We believe that such kinds of analyses can help in grasping the complex mechanisms of disorders, for which the interplay between bodily and psychological factors remains, for the most part, elusive. The current analysis provides a comprehensive assessment in a fairly large sample, and cross-sectional and longitudinal measures were evaluated. The calculations in our study should be interpreted as indications rather than precise estimations of caloric restriction; however, precisely estimating basal metabolic rate and physical activity was not the aim of the study. Future such studies would benefit from objective measures of caloric intake (e.g., via calorimetry) and of physical activity [59]. Another limitation of our exploratory analysis was the non-inclusion of information on the nutritional composition and preparation of foods ingested before admission and during the renutrition program. Furthermore, the use of tobacco was not registered, and the use of substances different from those in the exclusion criteria (e.g., stimulants of the glutamatergic type) was not assessed. Studies of the effects of nutrients on mood, anxiety, eating, and body image symptoms are still limited; however, EDs are increasingly conceptualized as disorders in which psychiatric and metabolic aspects are intertwined. Future studies should evaluate the effects of diet composition on ED pathology and recovery [24,27]. Furthermore, assessing the contribution of different body compositions (e.g., measuring the percentage of body fat) could most probably provide valuable insights. Lastly, evaluating the duration of caloric restriction, its quality (prolonged vs. intermittent), and its connections with other weight-related measures such as the rate of weight loss would provide a more detailed overview [10].

In conclusion, the results from this study suggest that restriction in caloric intake bears a positive relationship with body dissatisfaction at admission to inpatient treatment for ED, with weight suppression as a moderator in the relationship. Levels of caloric restriction still present at the end of therapy appear to be significantly influenced by high levels of perfectionistic concern over mistakes and state anxiety. Expanding the measures used in ED assessment can provide valuable insights into understanding the complex interplay between bodily and psychological features.

## Figures and Tables

**Figure 1 nutrients-15-03409-f001:**
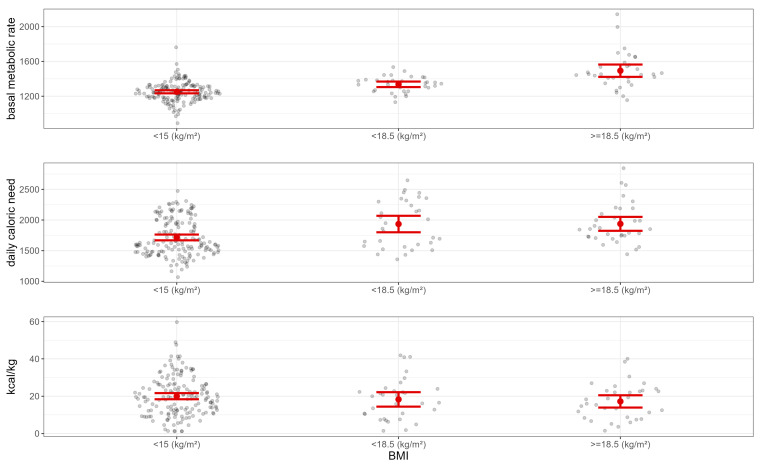
Basal metabolic rate (kcal/day), daily caloric need (kcal/day), and caloric intake (kcal/kg) for inpatients with body mass index (BMI) below 15, between 15 and 18.5, and above 18.5 kg/m^2^. Error lines represent standard error.

**Figure 2 nutrients-15-03409-f002:**
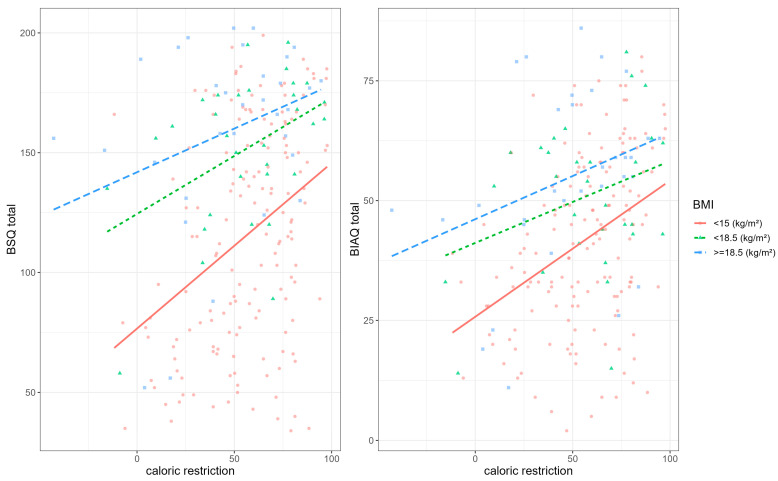
Relationship between caloric restriction and body shape questionnaire (BSQ) and body avoidance questionnaire (BIAQ) total for inpatients with body mass index (BMI) below 15, between 15 and 18.5, and above 18.5 kg/m^2^.

**Figure 3 nutrients-15-03409-f003:**
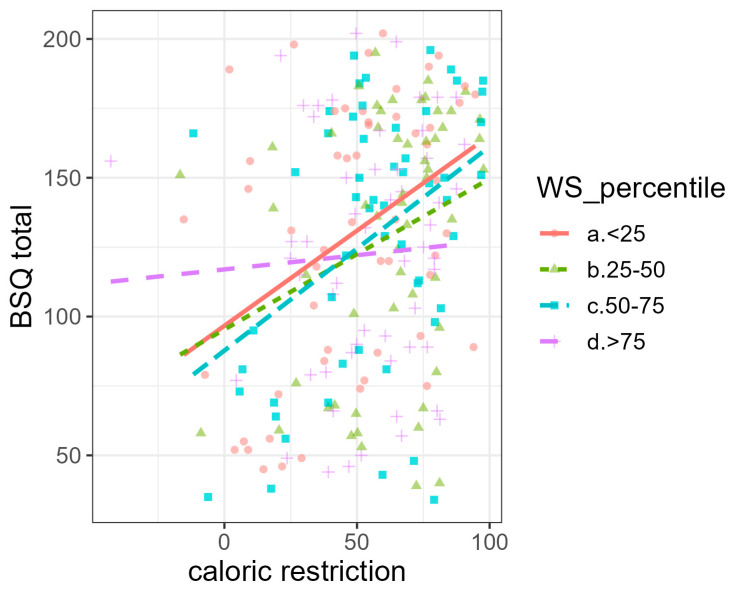
Relationship between caloric restriction and body shape questionnaire (BSQ) total in the whole sample for individuals with weight suppression (WS) (a) below the 25th percentile, (b) between the 25th and 50th percentile, (c) between the 50th and 75th percentile, and (d) above the 75th percentile. In the sample, the 25th percentile corresponds to 4.4 kg/m^2^, the 50th percentile corresponds to 6.3 kg/m^2^, and the 75th percentile corresponds to 8.73 kg/m^2^.

**Table 1 nutrients-15-03409-t001:** Clinical and sociodemographic characteristics of the sample.

Characteristic	N = 225 ^1^
Age (years)	25 (10)
Sex	
Female	210 (93%)
Male	15 (6.7%)
Education (years)	13.38 (2.90)
Student	127 (56%)
Occupation	150 (67%)
Ethnicity	
Hispanic	1 (0.4%)
Other	1 (0.4%)
White/Caucasian	223 (99%)
BMI	15.5 (4.5)
diagnosis	
AN-R	134 (60%)
AN-BP	51 (23%)
BN	20 (8.9%)
BED	3 (1.3%)
ARFID	8 (3.6%)
OSFED	8 (3.6%)
Duration of illness (years)	7 (9)
Binge-purging symptoms	84 (37%)
Psychiatric comorbidity	122 (54%)
Weight category	
BMI < 15	126 (56%)
BMI 15–15.99	34 (15%)
BMI 16–16.99	21 (9.3%)
BMI 17–18.49	11 (4.9%)
BMI 18.5–25	26 (12%)
BMI > 25	7 (3.1%)
History of overweight/obesity	40 (18%)
Weight suppression (kg/m^2^)	7.4 (5.8)
kcal/kg	19 (11)
BMR	1299 (154)
Daily calorie need	1779 (332)
Activity	
Sedentary	119 (53%)
Light	39 (17%)
Moderate	20 (8.9%)
Strong	47 (21%)
Caloric restriction ^2^	54 (28)
EDE-Q restraint	3.42 (2.10)
EDE-Q eating concern	3.17 (1.67)
EDE-Q shape concern	4.22 (1.71)
EDE-Q weight concern	3.74 (1.83)
EDE-Q global score	3.64 (1.69)
BSQ	127 (46)
BIAQ clothing	20 (10)
BIAQ social activities	9 (6)
BIAQ eating-related control behavior	7.2 (4.6)
BIAQ grooming/weighing	8.5 (3.6)
BIAQ total score	45 (19)
FMPS concern over mistakes	30 (10)
FMPS personal standards	24 (7)
FMPS parental expectations	11.0 (5.4)
FMPS parental criticism	9.9 (4.4)
FMPS doubts about actions	13.1 (4.1)
FMPS organization	23.9 (5.1)
STAI state anxiety	57 (14)
STAI trait anxiety	59 (13)
BDI total score	17 (8)

^1^ Mean (SD); n (%). ^2^ Caloric restriction is calculated as the percentage reduction in daily caloric intake from the individual’s daily caloric need. Abbreviations: AN-R = anorexia nervosa—restricting type; AN-BP = anorexia nervosa—binge-purging type; ARFID = avoidant/restrictive food intake disorder; BN = bulimia nervosa; OSFED = other specified feeding and eating disorder; BMI = body mass index; BMR = basal metabolic rate; EDE-Q = Eating Disorder Examination Questionnaire; STAI = State-Trait Anxiety Inventory; BDI = Beck Depression Inventory; BSQ = Body Shape Questionnaire; BIAQ = Body Image Avoidance Questionnaire; FMPS = Frost Multidimensional Perfectionism Scale; SD = standard deviation.

**Table 2 nutrients-15-03409-t002:** Admission and discharge variables in the underweight sample.

Characteristic	Admission, N = 192 ^1^	Discharge, N = 192 ^1^	*p*-Value ^2^
Age (years)	24 (10)	-	
Sex			
Female	180 (94%)	-	
Male	12 (6.3%)	-	
Diagnosis			
AN-R	134 (70%)	-	
AN-BP	51 (27%)	-	
ARFID	7 (3.6%)	-	
Duration of illness (years)	6 (9)	-	
Binge-purging symptoms	60 (31%)	-	
History of overweight/obesity	24 (13%)	-	
Weight suppression (kg/m^2^)	7.8 (5.5)	-	
Length of stay (days)	34 (16)	-	
Enteral therapy	51 (28%)	-	
Delta kcal ^3^	808 (471)	-	
kcal/week ^4^	189 (143)	-	
Weight (kg)/week	0.38 (0.47)	-	
BMI	14.21 (1.78)	14.84 (1.69)	<0.001
BMR	1265 (114)	1283 (116)	0.13
Daily calorie need	1752 (326)	1679 (290)	0.021
kcal/kg	20 (11)	40 (10)	<0.001
Caloric restriction ^5^	56 (25)	6 (26)	<0.001

^1^ Mean (SD); n (%). ^2^ Fisher’s exact test; Welch two-sample *t*-test. ^3^ Total increase in caloric intake (kcal/day) during the inpatient stay. ^4^ Mean weekly increase in caloric intake (kcal/day) during the inpatient stay. ^5^ Caloric restriction is calculated as the percentage reduction in daily caloric intake from the individual’s daily caloric need. Abbreviations: AN-R = anorexia nervosa—restricting type; AN-BP = anorexia nervosa—binge-purging type; ARFID = avoidant/restrictive food intake disorder; BMI = body mass index; BMR = basal metabolic rate.

**Table 3 nutrients-15-03409-t003:** Regression model with BSQ score as dependent variable.

Characteristic	Beta	95% CI ^1^	*p*-Value
Caloric restriction	19	13, 25	<0.001
Weight suppression (kg/m^2^)	−0.38	−6.0, 5.2	0.9
BMI	19	13, 26	<0.001
Binge-purging symptoms	18	6.1, 30	0.003
Age (years)	4.4	−5.9, 15	0.4
Duration of illness (years)	−5.6	−16, 4.5	0.3
Caloric restriction × weight suppression (kg/m^2^)	−7.5	−13, −1.8	0.010

^1^ CI, confidence interval. R^2^ of the model was 0.32.

**Table 4 nutrients-15-03409-t004:** Regression model with BIAQ score as dependent variable.

Characteristic	Beta	95% CI ^1^	*p*-Value
Caloric restriction	7.3	4.8, 9.9	<0.001
Weight suppression (kg/m^2^)	−0.07	−2.4, 2.3	>0.9
BMI	6.2	3.5, 8.8	<0.001
Binge-purging symptoms	4.2	−1.0, 9.4	0.11
Age (years)	0.13	−4.3, 4.5	>0.9
Duration of illness (years)	−1.3	−5.6, 3.1	0.6
Caloric restriction × weight suppression (kg/m^2^)	−3.1	−5.5, −0.70	0.012

^1^ CI, confidence interval. R^2^ of the model was 0.24.

**Table 5 nutrients-15-03409-t005:** Regression model with caloric intake at discharge as dependent variable.

Characteristic	Beta	95% CI ^1^	*p*-Value
BMR	−2.7	−4.1, −1.3	<0.001
Length of stay (days)	2.3	0.98, 3.7	<0.001
BMI	−1.1	−2.4, 0.16	0.086
Daily calorie need	−0.19	−1.6, 1.2	0.8
FMPS concern over mistakes	−2.6	−3.8, −1.4	<0.001

^1^ CI, confidence interval. R^2^ of the model was 0.33.

**Table 6 nutrients-15-03409-t006:** Regression model with caloric restriction at discharge as dependent variable.

Characteristic	Beta	95% CI ^1^	*p*-Value
Daily calorie need	5.2	1.1, 9.4	0.014
FMPS concern over mistakes	4.9	0.98, 8.8	0.014
STAI state anxiety	5.1	1.1, 9.0	0.012
BMR	3.2	−0.40, 6.8	0.081
EDE-Q global score	1.7	−2.6, 5.9	0.4

^1^ CI confidence interval. R^2^ of the model was 0.32.

## Data Availability

Data are not publicly available.

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
