# Peer review of "Exploring Caloric Restriction in Inpatients with Eating Disorders: Cross-Sectional and Longitudinal Associations with Body Dissatisfaction, Body Avoidance, Clinical Factors, and Psychopathology"

_nutrients, 2023, doi:10.3390/nu15153409_

Round 1
Reviewer 1 Report
In this study Martini et al. aim to investigate the possible correlation between caloric restriction and its relationship with psychopathological, clinical and anamnestic factors in individuals with ED, particularly AN patients. The question is of relevance for the field however data presentation and discussion are not clear.
Please, plot the data in the table to help with data visualization. Please, in the discussion elaborate more on the results and singularly examine them.
The results of this study might be innovative if the impact of caloric restriction is independent of initial body weight and/or final body weight (at the end of the remission).
Not clear from the data.
Minor mistakes. Some sentences of the discussion are too long.
Author Response
In this study Martini et al. aim to investigate the possible correlation between caloric restriction and its relationship with psychopathological, clinical and anamnestic factors in individuals with ED, particularly AN patients. The question is of relevance for the field however data presentation and discussion are not clear.
Please, plot the data in the table to help with data visualization. Please, in the discussion elaborate more on the results and singularly examine them.
Thank you for the useful suggestion. We have produced Figure 1, 2, and 3. Figure 1 shows data on basal metabolic rate (kcal/die), daily caloric need (kcal/die), and caloric intake (kcal/kg) at admission for different BMI groups. Figure 2 and 3 shows the relationship between caloric restriction, BSQ and BIAQ total respectively for different BMI groups and according to different weight suppression levels.
We have rewritten and expanded several parts of the discussion.
For instance,
“The mean CR in our sample was over 50%. Even though for underweight individuals the use of a formula for basal metabolic rate could have led to overestimations, this value is well above the range used in treatments with CR, thus limiting the possibility of comparisons with studies involving non-ED obese and normal-weight individuals. In our clinical sample, in line with our expectations, CR emerged as relevantly related to ED features across the weight and ED spectrum. …”;
“Interestingly, the expected association between CR and more severe body dissatisfaction and avoidance was independent from BMI (i.e., present across the weight spectrum). These findings suggest that the percentage of CR at admission to inpatient treatment could complement weight related indicators such as BMI, whose connection to psychopathology is limited [51]. “;
“The analyses conducted at the end of inpatient stay for underweight individuals highlight the relevance of perfectionistic concern over mistakes as a baseline predictor…Similarly, baseline anxiety symptoms were associated with higher caloric restriction at discharge, whereas neither specific eating symptoms nor admission or discharge BMI showed prominent roles in these associations. “
“Nonetheless, bidirectional associations between caloric restriction and psychopathology in individuals with EDs should be considered and assessed in future studies, with a focus both on quantitative restrictions and the nutritional content of foods.”
We believe that now the discussion is overall clearer and reflects the results described in previous sections.
The results of this study might be innovative if the impact of caloric restriction is independent of initial body weight and/or final body weight (at the end of the remission).
Thank you for raising this relevant point. Regarding regression models with BSQ and BIAQ as dependent variables, both BMI and caloric restriction are among the significant predictors. In these models, multicollinearity as measured with VIF in all predictors is low (i.e., < 5 as can be appreciated in supplementary figures). These results suggest that both BMI and caloric restriction are significantly and fairly independently related to BSQ an BIAQ total. We have highlighted these points in the text and added figure 2.
Regarding regression models assessing baseline predictors of caloric intake and restriction at discharge, we have re-run random forest including both admission and discharge BMI (all variables included in the random forest can be seen in supplementary table). This inclusion has not changed the best predictors for caloric restriction, whereas admission BMI has been included in the model regarding caloric intake, however not emerging as a significant predictor in the regression. If the reviewer deems it necessary we can conduct additional analyses.
Not clear from the data.
we hope that now independence of BMI and caloric restriction emerges from expanded text regarding regressions results discussion and figure 2 showing comparably increase in levels of BSQ and BIAQ corresponding to higher caloric restriction for different BMI levels.
Comments on the Quality of English Language
Minor mistakes. Some sentences of the discussion are too long.
We have re-worked the discussion section, divided some sentences, and corrected minor English mistakes.
Reviewer 2 Report
Nutrients 2509313
These results do not suggest that caloric restriction is a good indicator of eating disorder severity and do not warrant a multidimensional assessment of eating disorder psychopathology until:
The introduction includes the influence of calories from Maillard-reaction end-products (MEs) in baked, barbequed, broiled, dried, fried, grilled, irradiated, pressure-cooked, and sauteed nutrients on oxidative stress, immune-driven inflammation, mitochondria, differentiated and undifferentiated stem cells, neuropsychiatric and eating disorder phenotypes.
The introduction includes the influence of calories from Maillard-reaction intermediate products (MIPs) in pasteurized dairy and juices on redox imbalance, immune-driven inflammation, mitochondria, differentiated and undifferentiated stem cells, neuropsychiatric and eating disorder phenotypes.
The introduction includes the influence of calories from Maillard-reaction end-product-free (ME-free) raw, steamed, fondue, boiled, and normal pressure stewed meals, desserts, snacks, and beverages on redox balance, immune fortification, inflammation reversal, mitochondrial integrity, differentiated and undifferentiated stem cells stability, and reduction in the incidence of neuropsychiatric and eating disorder phenotypes.
The method and results include the approximated relative percentage of ME, MIP, and ME-free calories consumed daily for each subject studied.
The reference section is correspondingly populated with pertinent peer reviewed articles on advanced glycation end-products, advanced lipoxidation end-products, and their nucleic acid and protein relatives.
Then, and only then, is a justifiable suggestion made regarding caloric restriction and its indicator strength in each type of eating disorder after glutamate, alcohol, nicotine, and other substance use disorders have been ruled out.
The method and discussion therefore must include the DSM-5 differential diagnosis mandate. An eating disorder diagnosis cannot be made until after substance use disorders are ruled out.
Until then page 3 of 14, lines 113-115 ‘Exclusion criteria for this study were age <18 years old, presence of psychotic or active substance use disorder, or incomplete questionnaires on eating symptoms and body image’ may be an incomplete, unsubstantiated, and inaccurate differential diagnosis exclusion claim for stimulant use disorder, glutamate dependence, glutamatergic/dopaminergic type.
Quality of English Language Resulted in 6 Suggestions Made in this Small Sample:
In conclusion, the results from this study suggest that restriction in caloric intake bears a positive relationship with body dissatisfaction at admission to inpatient treatment for ED, with weight suppression as a moderator in the relationship. Levels of caloric restriction still present at the end of therapy appear to be significantly influenced by high levels of perfectionistic concern over mistakes and state anxiety. Expanding the measures used in ED assessment can provide valuable insights into understanding the complex interplay between bodily and psychological features.

Quality of English Language Resulted in 6 Suggestions Made in this Small Sample:
In conclusion, the results from this study suggest that restriction in caloric intake bears a positive relationship with body dissatisfaction at admission to inpatient treatment for ED, with weight suppression as a moderator in the relationship. Levels of caloric restriction still present at the end of therapy appear to be significantly influenced by high levels of perfectionistic concern over mistakes and state anxiety. Expanding the measures used in ED assessment can provide valuable insights into understanding the complex interplay between bodily and psychological features.
Author Response
These results do not suggest that caloric restriction is a good indicator of eating disorder severity and do not warrant a multidimensional assessment of eating disorder psychopathology until:
The introduction includes the influence of calories from Maillard-reaction end-products (MEs) in baked, barbequed, broiled, dried, fried, grilled, irradiated, pressure-cooked, and sauteed nutrients on oxidative stress, immune-driven inflammation, mitochondria, differentiated and undifferentiated stem cells, neuropsychiatric and eating disorder phenotypes.
The introduction includes the influence of calories from Maillard-reaction intermediate products (MIPs) in pasteurized dairy and juices on redox imbalance, immune-driven inflammation, mitochondria, differentiated and undifferentiated stem cells, neuropsychiatric and eating disorder phenotypes.
The introduction includes the influence of calories from Maillard-reaction end-product-free (ME-free) raw, steamed, fondue, boiled, and normal pressure stewed meals, desserts, snacks, and beverages on redox balance, immune fortification, inflammation reversal, mitochondrial integrity, differentiated and undifferentiated stem cells stability, and reduction in the incidence of neuropsychiatric and eating disorder phenotypes.
we have removed from the abstract and throughout the text the term "severity". We stress the exploratory nature of our study focused on the relationship between the quantity of calories ingested and eating disorders features. Nonetheless, we have expanded the introduction section to include relevant studies on nutritional composition of foods and advanced glycation end products in eating disorders:
“Despite the sheer quantity of calories introduced, diet composition could both impact and be influenced by health and psychopathology in individuals with EDs. Furthermore, cooking methods influence nutritional qualities and the content of potentially dangerous compounds present in foods, such as advanced glycation end-products (AGEs), which are associated with metabolic disease and could be involved in gut microbiome dysfunction [19,20]. For instance, even though 10–30% of AGEs introduced with foods are absorbed, a diet rich in AGEs could be involved in maintaining obesity [20]. On the opposite weight polarity, due to AN psychopathology, the diet of individuals with AN is characterized by products generally considered as healthy and with low levels of AGEs (i.e., low-fat foods, foods cooked at low temperatures, raw foods) [19,21]. Still, paradoxically higher plasma AGEs levels have been found in individuals with AN in comparison to controls, suggesting increased endogenous production of these compounds, coupled with concomitant oxidative dysfunction [22]. Given the conceptualization of EDs as metabo-psychiatric disorders [23], the increasing importance recognized in the ED field to the role of the gut microbiome [24], immunity [25], and inflammation [26], more attention should be given to the nutritional and metabolic effects of foods chosen by patients and introduced during renutrition [24,27]. Associations between diet composition and depressive and anxiety symptoms have emerged from the literature [28], however similar studies in the ED field are still lacking. If significant associations between caloric restriction in EDs and clinical factors emerged, future studies should evaluate the effects of nutritional composition, metabolic, and inflammatory actions of food ingested in this relation.”
The method and results include the approximated relative percentage of ME, MIP, and ME-free calories consumed daily for each subject studied.
Unfortunately, we are unable to provide estimation regarding these products. We have higlighted this limitation in the discussion section.
“Another limitation of our exploratory analysis is the non-inclusion of information on the nutritional composition and preparation of foods ingested before admission and during the renutrition program… The study of the effects of nutrients on mood, anxiety, eating, and body image symptoms is still limited, however, EDs are increasingly conceptualized as disorders in which psychiatric and metabolic aspects are intertwined. Future studies should evaluate the effects of diet composition on ED pathology and recovery [24,27].”
The reference section is correspondingly populated with pertinent peer reviewed articles on advanced glycation end-products, advanced lipoxidation end-products, and their nucleic acid and protein relatives.
we have introduced the following references on the topic. If the reviewer has other suggested references, we can include them.
- Zawada, A.; Machowiak, A.; Rychter, A.M.; Ratajczak, A.E.; Szymczak-Tomczak, A.; Dobrowolska, A.; Krela-Kaźmierczak, I. Accumulation of Advanced Glycation End-Products in the Body and Dietary Habits. Nutrients 2022, 14, 3982.
- Bettiga, A.; Fiorio, F.; Di Marco, F.; Trevisani, F.; Romani, A.; Porrini, E.; Salonia, A.; Montorsi, F.; Vago, R. The Modern Western Diet Rich in Advanced Glycation End-Products (AGEs): An Overview of Its Impact on Obesity and Early Progression of Renal Pathology. Nutrients 2019, 11, 1748.
- Misra, M.; Tsai, P.; Anderson, E.J.; Hubbard, J.L.; Gallagher, K.; Soyka, L.A.; Miller, K.K.; Herzog, D.B.; Klibanski, A. Nutrient Intake in Community-Dwelling Adolescent Girls with Anorexia Nervosa and in Healthy Adolescents. The American journal of clinical nutrition 2006, 84, 698–706.
- Kovalčı́ková, A.G.; Tichá, L.; Šebeková, K.; Celec, P.; Čagalová, A.; Sogutlu, F.; Podracká, L. Oxidative Status in Plasma, Urine and Saliva of Girls with Anorexia Nervosa and Healthy Controls: A Cross-Sectional Study. Journal of eating disorders 2021, 9, 1–12.
- Watson, H.J.; Yilmaz, Z.; Thornton, L.M.; Hübel, C.; Coleman, J.R.; Gaspar, H.A.; Bryois, J.; Hinney, A.; Leppä, V.M.; Mattheisen, M.; et al. Genome-Wide Association Study Identifies Eight Risk Loci and Implicates Metabo-Psychiatric Origins for Anorexia Nervosa. Nature genetics 2019, 51, 1207–1214.
- Herpertz-Dahlmann, B.; Seitz, J.; Baines, J. Food Matters: How the Microbiome and Gut–Brain Interaction Might Impact the Development and Course of Anorexia Nervosa. European child & adolescent psychiatry 2017, 26, 1031–1041.
- Butler, M.J.; Perrini, A.A.; Eckel, L.A. The Role of the Gut Microbiome, Immunity, and Neuroinflammation in the Pathophysiology of Eating Disorders. Nutrients 2021, 13, 500.
- Dalton, B.; Campbell, I.C.; Chung, R.; Breen, G.; Schmidt, U.; Himmerich, H. Inflammatory Markers in Anorexia Nervosa: An Exploratory Study. Nutrients 2018, 10, 1573.
- Pettersson, C.; Svedlund, A.; Wallengren, O.; Swolin-Eide, D.; Karlsson, G.P.; Ellegård, L. Dietary Intake and Nutritional Status in Adolescents and Young Adults with Anorexia Nervosa: A 3-Year Follow-up Study. Clinical Nutrition 2021, 40, 5391–5398.
- Aucoin, M.; LaChance, L.; Naidoo, U.; Remy, D.; Shekdar, T.; Sayar, N.; Cardozo, V.; Rawana, T.; Chan, I.; Cooley, K. Diet and Anxiety: A Scoping Review. Nutrients 2021, 13, 4418.
Then, and only then, is a justifiable suggestion made regarding caloric restriction and its indicator strength in each type of eating disorder after glutamate, alcohol, nicotine, and other substance use disorders have been ruled out.
The method and discussion therefore must include the DSM-5 differential diagnosis mandate. An eating disorder diagnosis cannot be made until after substance use disorders are ruled out.
Until then page 3 of 14, lines 113-115 ‘Exclusion criteria for this study were age <18 years old, presence of psychotic or active substance use disorder, or incomplete questionnaires on eating symptoms and body image’ may be an incomplete, unsubstantiated, and inaccurate differential diagnosis exclusion claim for stimulant use disorder, glutamate dependence, glutamatergic/dopaminergic type.
We have modified the sentences on exclusion criteria as follows:
“Exclusion criteria for this study were age <18 years old, presence of psychotic disorders, presence of active alcohol, cannabis, hallucinogens, inhalants, opioids, sedatives, cocaine, amphetamines, or methamphetamine use disorder. Finally, only individuals with complete questionnaires on eating symptoms and body image were included.”
We assess the presence of these disorders for clinical and research purposes, even though their exclusion is not required in the DSM-5 anorexia nervosa, bulimia nervosa, binge-eating disorder, and other specified feeding or eating disorder criteria.
In the limitation section we now specify that we did not evaluate the number of individuals smoking tobacco, and that use of other substances not specified in the exclusion criteria was not specifically assessed: "Furthermore, the use of tobacco was not registered, and the use of substances different from those in the exclusion criteria (e.g., stimulants of the glutamatergic type) was not assessed."
Quality of English Language Resulted in 6 Suggestions Made in this Small Sample:
In conclusion, the results from this study suggest that restriction in caloric intake bears a positive relationship with body dissatisfaction at admission to inpatient treatment for ED, with weight suppression as a moderator in the relationship. Levels of caloric restriction still present at the end of therapy appear to be significantly influenced by high levels of perfectionistic concern over mistakes and state anxiety. Expanding the measures used in ED assessment can provide valuable insights into understanding the complex interplay between bodily and psychological features.
Thank you, we have modified this paragraph accordingly.